# School-Age Outcomes of Antenatal Magnesium Sulphate in Preterm Infants

**DOI:** 10.3390/children10081324

**Published:** 2023-07-31

**Authors:** Akira Kobayashi, Masato Ito, Erika Ota, Fumihiko Namba

**Affiliations:** 1General Center for Perinatal, Maternal and Neonatal Medicine, Niigata University Medical and Dental Hospital, Niigata 951-8520, Japan; 2Department of Pediatrics, Akita University Graduate School of Medicine, Akita 010-8543, Japan; i1t2o3u4jp@gmail.com; 3Global Health Nursing, Graduate School of Nursing Science, St. Luke’s International University, Chuo-ku, Tokyo 104-0044, Japan; ota@slcn.ac.jp; 4Tokyo Foundation for Policy Research, Minato-ku, Tokyo 106-0032, Japan; 5Department of Pediatrics, Saitama Medical Center, Saitama Medical University, Saitama 350-8550, Japan

**Keywords:** magnesium sulfate, preterm infants, outcome assessment

## Abstract

Background: Antenatal magnesium sulphate (MgSO_4_) therapy given to women at risk of preterm birth reduced the risk of cerebral palsy in early childhood. However, its effect on longer-term neurological outcomes remains uncertain. This study aimed to assess the effects of antenatal MgSO_4_ therapy on school-age outcomes of preterm infants. Methods: We conducted a systematic review and meta-analysis. We searched MEDLINE, EMBASE, CENTRAL, and CINAHL for randomized controlled trials (RCTs). Two reviewers independently evaluated the eligibility for inclusion and extracted data. Results: Ten RCTs were included. Only two of them were on school-age outcomes. Antenatal MgSO_4_ therapy had no impact on cerebral palsy, hearing impairment, neurosensory disability, and death at school-age. Meta-analysis on mental retardation and visual impairment was not able to be performed due to different methods of evaluation. In the analysis of short-term outcomes conducted as secondary outcomes, antenatal MgSO_4_ therapy increased the risk of maternal adverse events with any symptom (3 RCTs; risk ratio 2.79; 95% confidence interval 1.10 to 7.05, low certainty of evidence) but was not associated with any neonatal symptoms. Conclusions: The number of cases was insufficient to determine the impact of antenatal MgSO_4_ therapy on school-age outcomes. Further accumulation of long-term data is required.

## 1. Introduction

Preterm birth is the most common cause of perinatal and infant mortality [1]. Preterm infants who survive have a higher risk for neurological impairments such as cerebral palsy (CP), cognitive dysfunction, and visual and hearing impairments. In consequence, they face an increased likelihood of substantial disability and developmental delays [2].

Antenatal magnesium sulphate (MgSO_4_) is used for maternal eclampsia and protection of premature delivery [3,4]. There have been some reports since the 1990s that antenatal MgSO_4_ therapy has an effect of reduction in CP [5,6,7,8,9,10]. It is presumed that MgSO_4_ offers neuroprotective effects through mechanisms such as inhibiting the entry of calcium into the fetal brain, sustaining the balance of glutamate, and diminishing the interaction with oxygen-free radicals in the fetus [11]. As a result, the Cochrane review published in 2009 revealed that antenatal MgSO_4_ therapy given to women at risk of preterm birth significantly reduced the risk of CP in early childhood (risk ratio (RR) 0.68; 95% confidence interval (CI) 0.54 to 0.87) [12]. Two subsequent meta-analyses, an individual participant data meta-analysis [13] and a systematic review with meta-analysis and trial sequential analysis [14], showed similar results. However, no significant effect on short-term neurological outcomes other than CP was found. In addition, the effects of antenatal MgSO_4_ therapy on longer-term neurological outcomes are still unknown.

Therefore, in this systematic review, we aimed to evaluate the effects of antenatal MgSO_4_ therapy given to women at risk of preterm birth on school-age outcomes of preterm infants.

## 2. Methods

We conducted a systematic review and meta-analysis according to the Preferred Reporting Items for Systematic Reviews and Meta-Analyses (PRISMA) statement [15]. The protocol was registered on PROSPERO, the international prospective register of systematic reviews (registration number: CRD 42017064070).

### 2.1. Inclusion Criteria

We included individual and cluster randomized controlled trials (RCTs) comparing antenatal MgSO_4_ therapy to placebo for women at risk of preterm birth.

### 2.2. Search Strategy

We searched MEDLINE, EMBASE, CENTRAL, CINAHL, and any other accessible, relevant databases on October 22, 2020, with a publication restriction of 2008–2020, but no language or document type limitations. We limited the period of literature search to 2008–2020 because we used the results of the Cochrane review published in 2009, in which the literature before 2008 had been searched. The selection of keywords was accomplished by incorporating expert opinions, conducting a comprehensive literature review, utilizing controlled vocabulary such as Medical Subject Headings and Excerpta Medica Tree, and thoroughly examining the primary search results. The final search terms included “magnesium sulfate”, “prenatal care” (Appendix A).

### 2.3. Identification of Studies

The search activities were carried out by an information specialist, while the reviewers manually conducted supplemental searches in only MEDLINE. To gather information on unpublished trials, the authors were directly contacted for additional details. Two of the authors involved in this review independently evaluated all the studies identified for further analysis. We defined the inclusion criteria as the following: Population; women at risk of preterm birth, intervention; antenatal MgSO_4_ therapy, control; placebo, study design; RCTs. First, we checked titles and abstracts of articles and excluded articles we decided were unnecessary. Second, we read the full-text articles we had adopted and excluded articles we decided were unnecessary. If there was a disagreement between two reviewers during this process, each presented the reasons for inclusion or exclusion and resolved through discussion. When consensus was not reached, we consulted with a third evaluator and followed his decision. Piloted data extraction forms were employed by two of the reviewers to gather fundamental information regarding the studies, including participant details, descriptions of treatment and control interventions, as well as outcome data; (1) school-age outcomes (6–14 years old) including CP, mental retardation, visual impairment, hearing impairment, neurosensory disability, and death, (2) short-term outcomes including (a) extended duration of the pregnancy, (b) serum magnesium concentration of mother, cord blood, and newborn on admission, (c) maternal adverse events, (d) neonatal adverse events, (e) small for gestational age (SGA), (f) use of ventilator or vasopressor, (g) morbidity including neonatal asphyxia, respiratory distress syndrome (RDS), bronchopulmonary dysplasia (BPD), patent ductus arteriosus (PDA), intraventricular hemorrhage (IVH), periventricular leukomalacia (PVL), seizure, sepsis, necrotizing enterocolitis (NEC), and retinopathy of prematurity (ROP), (h) length of stay in hospital or neonatal intensive care unit.

### 2.4. Data Analysis

The data analysis was performed utilizing Review Manager 5.4 software (RevMan 5.4, 2014 “http://community.cochrane.org/tools/review-production-tools/revman-5 (accessed on 21 July 2017)”). For dichotomous data, RR was employed, while for continuous data, either the weighted mean difference or standardized mean difference was utilized. A fixed-effects meta-analysis was conducted to analyze the data, and the outcomes are presented as the average intervention effect along with 95% CIs. In cases where high heterogeneity was observed, a random-effects model was employed. The I^2^ test was utilized to estimate the degree of statistical heterogeneity. A value of I^2^ greater than 75% indicates substantial heterogeneity, while a range of 30% to 60% may suggest moderate heterogeneity. We conducted analysis, as far as possible, on an intention-to-treat basis. In each trial, the denominator for each outcome was the randomized number minus the number of missing participants. Sensitivity analyses were conducted to explore the impact of employing either fixed-effects or random-effects analyses for outcomes displaying statistical heterogeneity. We did not conduct subgroup analysis.

### 2.5. Assessment of Risk of Bias in Included Studies

Two of the authors independently evaluated the risk of bias 1.0 in the included studies using Cochrane risk of bias from the Cochrane Handbook for Systematic Reviews of Interventions [16]. We assessed seven domains: random sequence generation, allocation concealment, blinding of participants and personnel, blinding of outcome assessment, incomplete outcome data, selective reporting bias, and other biases. To address incomplete outcome data, we examined the reporting of attrition and dropouts, as well as the number of participants included in the analysis at each stage. We also investigated the reasons for attrition or exclusion when provided and assessed whether the missing data were evenly distributed among groups or correlated with the outcomes. Disagreements were resolved through discussion between the authors or by consulting with a third evaluator.

### 2.6. Certainty of Evidence

To assess the level of certainty in the evidence, we employed the summary of findings template from the Guideline Development Tool created by the Grading of Recommendations, Assessment, Development, and Evaluation (GRADE) Working Group (available at “http://www.gradeworkinggroup.org (accessed on 12 September 2020)”). The evaluation of the studies followed the GRADE guidelines, and the evidence was rated based on five criteria outlined by GRADE: risk of bias, inconsistency, indirectness, imprecision, and publication bias [17]. A certainty rating was given for CP, severe CP, death at school age, hearing impairment, neurosensory disability, and maternal adverse events (any symptom). The ratings obtained were categorized based on the four levels of certainty recommended by the GRADE approach, namely high, moderate, low, and very low levels of evidence certainty.

## 3. Results

### 3.1. Search Results

Database searches identified 535 reports. Of these, 48 reports were removed by duplicate, 426 reports were excluded by title and abstract screening, and 55 reports were excluded by full-text screening. Six reports (Doyle 2014 [18], Chollat 2014 [19], Paradidis 2012 [20], Drassinower 2015 [21], Colon 2016 [22], and Wolf 2020 [23]) met the inclusion criteria. In order to use the literature before 2008, we also added five reports (Crowther 2003 [7], Marret 2006 [8], Mittrendorf 2002 [24], Rouse 2008 [10], and Magpie 2006 [25]) adopted for analysis in Cochrane review published in 2009 [12]. We excluded Magpie 2006 because data from pregnant women less than 37 weeks gestation were not available even though we tried to contact the authors to obtain them. Finally, 10 reports were included in this analysis (Figure 1).

### 3.2. Included Studies

Ten studies were included in this review. Of them, eight studies were multi-center trials, and two studies (Mittrendorf 2002 and Paradidis 2012) were single-center trials. Mittrendorf 2002 had two treatment strategies. In one strategy (tocolytic arm), the treatment group was given antenatal MgSO_4_, whereas the control group was given other tocolytic agents such as ritodrine. Therefore, this strategy (tocolytic arm) was excluded from this review. Some differences existed in participants, interventions, and outcomes between studies. Regarding outcomes, two studies (Doyle 2014 and Chollat 2014) reported school-age outcomes, and eight other papers reported short-term outcomes. In addition, Doyle 2014 and Chollat 2014 were studies of school-age outcomes of Crowther 2003 and Marret 2006, respectively (Table 1).

### 3.3. Participants

Participants in the studies were pregnant women less than 37 weeks gestation (Table 2).

### 3.4. Interventions

The administration methods of MgSO_4_ were different in each study (Table 3).

### 3.5. Outcomes

School-age outcomes were evaluated in two studies (Doyle 2014 and Chollat 2014). Short-term outcomes were evaluated in eight other studies. Regarding school-age outcomes, in Doyle 2014, the outcomes were mortality, CP, motor function, general intellectual ability, academic skills, attention, executive function, behavior, growth, and functional and other neurosensory outcomes. They were evaluated between the modified 6–11 years. In Chollat 2014, the outcomes were a composite of death and motor outcome, composites of learning/cognitive disabilities and special education services, behavioral and psychiatric disorders, and health and neurosensory deficits.

### 3.6. Risk of Bias

Assessment of the risk of bias 1.0 of the 10 studies was conducted (Figure 2).

#### 3.6.1. Random Sequence Generation

Random sequence was generated by computer in the nine studies. In Mittrendorf 2002, the randomization method was unknown.

#### 3.6.2. Allocation Concealment

Central allocation was conducted in the nine studies. In Mittrendorf 2002, the allocation method was unknown.

#### 3.6.3. Blinding

All participants were blinded. Blinding of medical staff was conducted in the eight studies, not conducted in the two studies (Marret 2006 and Chollat 2014), but the influence on outcome assessment seemed low.

#### 3.6.4. Incomplete Outcome Data

In Drassinower 2015, all participants were followed. In the eight studies, not all participants were followed. For example, in Doyle 2014, the dropout rate was 32.9% and 28.9% in the MgSO_4_ group and the placebo group, respectively. In Chollat 2014, the dropout rate was 28.4% and 25.3% in the MgSO_4_ group and the placebo group, respectively. In Mittrendorf 2002, the follow-up rate was unknown (Appendix A).

#### 3.6.5. Selective Reporting

Prespecified results were reported in all studies.

### 3.7. Certainty of Evidence (GRADE)

The results of this are summarized in Table 4. All of the outcomes were evaluated as low certainty of evidence. Certainty of evidence downgraded due to attrition bias was high, or wide 95%CI crossing non-significant line, or high heterogeneity.

### 3.8. Effects of the Interventions

#### 3.8.1. School-Age Outcomes

##### CP

Two studies (Doyle 2014 and Chollat 2014) reported on CP. The MgSO_4_ group was not significantly different from the placebo group (RR, 0.99; 95% CI 0.69–1.41, I^2^ = 0%; low certainty of evidence) (Figure 3A).

##### Severe CP

Two studies (Doyle 2014 and Chollat 2014) reported on this outcome. Doyle 2014 defined severe CP as “Gross Motor Function Classification System level 5 (has limited voluntary control of movement) or level 4 (uses a wheelchair). Chollat 2014 defined severe CP as “inability to walk or walking only with aid”. The MgSO_4_ group was not significantly different from the placebo group (RR, 0.81; 95% CI 0.34–1.92, I^2^ = 0%; low certainty of evidence) (Figure 3B).

##### Mental Retardation

Two studies (Doyle 2014 and Chollat 2014) reported on mental retardation. In Doyle 2014, the score of developmental tests such as intelligence quotient was reported. For example, the Mean full-scale IQ was 93.8 and 94.9 in the MgSO_4_ group and the placebo group, respectively. No significant difference was found. On the other hand, in Chollat 2014, the number of people who recognized cognitive impairment was reported. Cognitive deficits/learning disabilities (moderate and severe) were 139/218 (63.8%) and 137/211 (64.9%) in the MgSO_4_ group and the placebo group, respectively. No significant difference was found. We were not able to conduct a meta-analysis because the evaluation methods were different in the two studies.

##### Visual Impairment

Two studies (Doyle 2014 and Chollat 2014) reported on visual impairment. In Doyle 2014, the presence or absence of a blind was reported. Blindness was 1/269 (0.4%) and 0/285 (0%) in the MgSO_4_ group and the placebo group, respectively. No significant difference was found. On the other hand, in Chollat 2014, the presence or absence of wearing glasses was reported. A visual deficiency was 105/218 (48.2%) and 98/211 (46.5%) in the MgSO_4_ group and the placebo group, respectively. No significant difference was found. We were not able to perform a meta-analysis because the evaluation methods were different in the two studies.

##### Hearing Impairment

Two studies (Doyle 2014 and Chollat 2014) reported on hearing impairment. The MgSO_4_ group was not significantly different from the placebo group (RR, 0.72; 95% CI 0.38–1.38, I^2^ = 0%; low certainty of evidence) (Figure 3C).

##### Neurosensory Disability

Two studies (Doyle 2014 and Chollat 2014) reported on neurosensory disability. The MgSO_4_ group was not significantly different from the placebo group (RR, 0.97, 95% CI 0.87–1.08, I^2^ = 32%; low certainty of evidence) (Figure 3D).

##### Death at School-Age

Two studies (Doyle 2014 and Chollat 2014) reported on this outcome. The MgSO_4_ group was not significantly different from the placebo group (RR, 0.83; 95% CI 0.68–1.02, I^2^ = 0%; low certainty of evidence) (Figure 3E).

#### 3.8.2. Short-Term Outcomes (Appendix A)

##### Maternal Adverse Events (Any Symptom)

Three studies (Crowther 2003, Rouse 2008, and Colon 2016) reported on this outcome. Maternal adverse events (any symptom) were significantly higher in the MgSO_4_ group (RR, 2.79; 95% CI 1.10–7.05, I^2^ = 98%; low certainty of evidence). We conducted sensitivity analysis and presented random-effects model results as the final model for statistical heterogeneity meta-analysis.

##### Maternal Adverse Events (Headache)

Two studies (Marret 2006 and Colon 2016) reported on this outcome. The MgSO_4_ group was not significantly different from the placebo group (RR, 3.59; 95% CI 0.60–21.54, I^2^ = 0%).

##### Maternal Adverse Events (Dizziness)

Two studies (Crowther 2003 and Colon 2016) reported on this outcome. Maternal dizziness was significantly higher in the MgSO_4_ group (RR, 2.22; 95% CI 1.54–3.20, I^2^ = 0%).

##### Neonatal Adverse Events (Any Symptom)

Two studies (Mittrendorf 2002 and Drassinower 2015) reported on this outcome. The MgSO_4_ group was not significantly different from the placebo group (RR, 0.96; 95% CI 0.89–1.03, I^2^ = 53%).

##### SGA

Two studies (Paradidis 2012 and Drassinower 2015) reported on SGA. The MgSO_4_ group was not significantly different from the placebo group (RR, 2.14; 95% CI 0.97–4.72, I^2^ = 0%).

##### Use of Ventilator

Five studies (Crowther 2003, Marret 2006, Rouse 2008, Colon 2016, and Wolf 2020) reported on this outcome. No significant difference was found between the MgSO_4_ group and the placebo group (RR, 0.93; 95% CI 0.82–1.07, I^2^ = 85%). We conducted sensitivity analysis and presented random-effects model results as the final model for statistical heterogeneity meta-analysis.

##### Use of Vasopressor

Three studies (Rouse 2008, Paradidis 2012, and Wolf 2020) reported on this outcome. No significant difference was found between the MgSO_4_ group and the placebo group (RR, 0.93; 95% CI 0.62–1.40, I^2^ = 74%). We conducted sensitivity analysis and presented random-effects model results as the final model for statistical heterogeneity meta-analysis.

##### Neonatal Asphyxia (5 Min Apgar Score < 7)

Three studies (Marret 2006, Rouse 2008, and Wolf 2020) reported on this outcome. The MgSO_4_ group was not significantly different from the placebo group (RR, 1.02; 95% CI 0.88–1.19, I^2^ = 14%).

##### RDS

Three studies (Marret 2006, Rouse 2008, and Colon 2016) reported on RDS. The MgSO_4_ group was not significantly different from the placebo group (RR, 0.97; 95% CI 0.90–1.05, I^2^ = 59%).

##### BPD

Three studies (Crowther 2003, Rouse 2008, and Wolf 2020) reported on BPD. The MgSO_4_ group was not significantly different from the placebo group (RR, 1.03; 95% CI 0.93–1.14, I^2^ = 44%).

##### PDA

Three studies (Rouse 2008, Paradidis 2012, and Wolf 2020) reported on PDA. The MgSO_4_ group was not significantly different from the placebo group (RR, 0.93; 95% CI 0.78–1.11, I^2^ = 0%).

##### IVH

Five studies (Mittrendorf 2002, Crowther 2003, Marret 2006, Rouse 2008, and Wolf 2020) reported on IVH. The MgSO_4_ group was not significantly different from the placebo group (RR, 0.94; 95% CI 0.84–1.05, I^2^ = 9%).

##### PVL

Five studies (Mittrendorf 2002, Crowther 2003, Marret 2006, Rouse 2008, and Wolf 2020) reported on PVL. The MgSO_4_ group was not significantly different from the placebo group (RR, 0.93; 95% CI 0.69–1.24, I^2^ = 0%).

##### Seizure

Three studies (Marret 2006, Rouse 2008, and Wolf 2020) reported on the seizure. The MgSO_4_ group was not significantly different from the placebo group (RR, 0.83; 95% CI 0.53–1.30, I^2^ = 0%).

##### Sepsis

Two studies (Rouse 2008 and Colon 2016) reported on sepsis. The MgSO_4_ group was not significantly different from the placebo group (RR, 0.97; 95% CI 0.81–1.15, I^2^ = 0%).

##### NEC

Four studies (Crowther 2003, Marret 2006, Rouse 2008, and Wolf 2020) reported on NEC. The MgSO_4_ group was not significantly different from the placebo group (RR, 1.21; 95% CI 0.98–1.50, I^2^ = 0%).

##### ROP

Two studies (Rouse 2008 and Wolf 2020) reported on ROP. The MgSO_4_ group was not significantly different from the placebo group (RR, 1.01; 95%CI 0.87–1.16, I^2^ = 0%).

## 4. Discussion

We conducted a systematic review and meta-analysis to evaluate the effects of antenatal MgSO_4_ therapy in women at risk of preterm birth on school-age outcomes and short-term outcomes in preterm infants. Antenatal MgSO_4_ therapy did not influence school-age outcomes, including CP, hearing impairment, neurosensory disability, and death at school age. Regarding short-term outcomes, the rate of any maternal adverse events, such as headache and dizziness, was significantly higher in the MgSO_4_ group. However, antenatal MgSO_4_ therapy was not associated with any short-term neonatal outcomes.

There have been no reports of meta-analysis on the effects of antenatal MgSO_4_ therapy for women at risk of preterm birth on school-age outcomes in preterm infants. Only two studies (Doyle 2014 [18] and Chollat 2014 [19]) have reported the association between antenatal MgSO_4_ therapy and school-age outcomes. The Cochrane review which was published in 2009, reported that antenatal MgSO_4_ therapy for women at risk of preterm birth for neuroprotection of the fetus reduced the risk of CP in early childhood [12]. However, we were not able to show the association between antenatal MgSO_4_ therapy and school-age outcomes such as CP. The reason would be because only 1038/1943 = 53.4% of the original participants were followed up to school age. In addition, out of 5 studies adopted in the Cochrane review in 2009, only 1 study (Rouse 2008 [10]) reported that moderate or severe CP occurred significantly less frequently in the MgSO_4_ group (RR 0.55; 95% CI 0.32–0.95). However, we had no school-age data from Rouse 2008. CP is unlikely to be diagnosed after two years of age and does not disappear once it has been diagnosed. Therefore, it is more reliable to draw conclusions from data collected in early childhood, which had a higher follow-up rate. The best evidence from infant studies shows that antenatal MgSO_4_ therapy has an impact on CP, but long-term data from school-age populations were insufficient to show the same. Accumulation of long-term data, including school-age, is strongly needed.

A meta-analysis of short-term outcomes was conducted by adding four new studies to the four studies adopted in the Cochrane Review published in 2009. Of the four added studies, Colon 2016 [22] and Wolf 2020 [23] were new RCTs. Paradidis 2012 [20] was a single-center analysis of Crowther 2003. Drassinower 2015 [21] was a secondary analysis of Rouse 2008. Seventeen items were able to be meta-analyzed in this systematic review. Of these, 12 items (use of a ventilator, use of vasopressor, neonatal asphyxia, RDS, BPD, PDA, IVH, PVL, seizure, sepsis, NEC, and ROP) were also reported in the previous reports [12,13,14], and the other 5 items (maternal adverse events with any symptom, headache, dizziness, neonatal adverse events with any symptom, and SGA) were firstly reported in this systematic review.

Regarding the former 12 items, the results of this review were not different from the previous reports. However, Wolf et al. reported that the risk of severe intraventricular hemorrhage (grade 3–4) underwent a borderline significant reduction in the MgSO_4_ groups (RR 0.77; 95% CI 0.60–1.00) [14]. In addition, the risk of intubation and/or chest compressions in the delivery room and the need for endotracheal intubation during the first hospitalization also underwent borderline significant reduction. A detailed analysis might show further validity. In the latter five items, maternal adverse events increased significantly in the MgSO_4_ group, but no serious adverse events, such as maternal death or use of a ventilator, were reported. There were no significant differences in short-term neonatal outcomes. However, these results were similar to previous results, so they were not so novel. The other items (extended duration of the pregnancy, serum magnesium concentration of mother, cord blood, and newborn on admission, length of stay in hospital or neonatal intensive care unit) were not able to be meta-analyzed because there have been no reports examining these items since 2008. However, a recent network meta-analysis evaluating tocolytic drugs reported that MgSO_4_ was probably effective in delaying preterm birth by 48 h (RR 1.12; 95% CI 1.02–1.23) [4]. Antenatal MgSO_4_ therapy may contribute to an improvement of outcomes for preterm infants, both by prolonging the gestational duration and by protecting the fetal brain.

There are several limitations in this review as follows; first, regarding school-age outcomes, the number of cases was insufficient because there were only two studies. We need to interpret with caution. If we have further reports on school-age outcomes, the results might change. Second, we were not able to conduct a meta-analysis on mental retardation and visual impairment due to the difference in evaluation methods. This would be solved by conducting an individual participant data meta-analysis. Third, considering short-term outcomes, we were not able to include Magpie 2006. And there is ongoing RCT of antenatal MgSO_4_ therapy [26]. These studies will provide more firm evidence.

In this systematic review, antenatal MgSO_4_ therapy in women at risk of preterm birth was not associated with the school-age outcomes of preterm infants. However, the number of cases was insufficient to provide evidence. On the other hand, the previous reports examined early childhood outcomes, which had a high follow-up rate, the Cochrane Review 2009, an individual participant data meta-analysis [13] and a systematic review with meta-analysis and trial sequential analysis [14], showed that antenatal MgSO_4_ therapy reduced the risk of CP in early childhood with no serious adverse events both on the mothers and newborns. Therefore, at present, antenatal MgSO_4_ therapy is recommended for women at imminent risk of preterm birth.

## 5. Conclusions

The number of cases was insufficient to determine the impact of antenatal MgSO_4_ therapy on school-age outcomes. Further accumulation of long-term data is required. Regarding short-term outcomes, antenatal MgSO_4_ therapy was associated with mild maternal adverse events but not with neonatal symptoms.

## Figures and Tables

**Figure 1 children-10-01324-f001:**
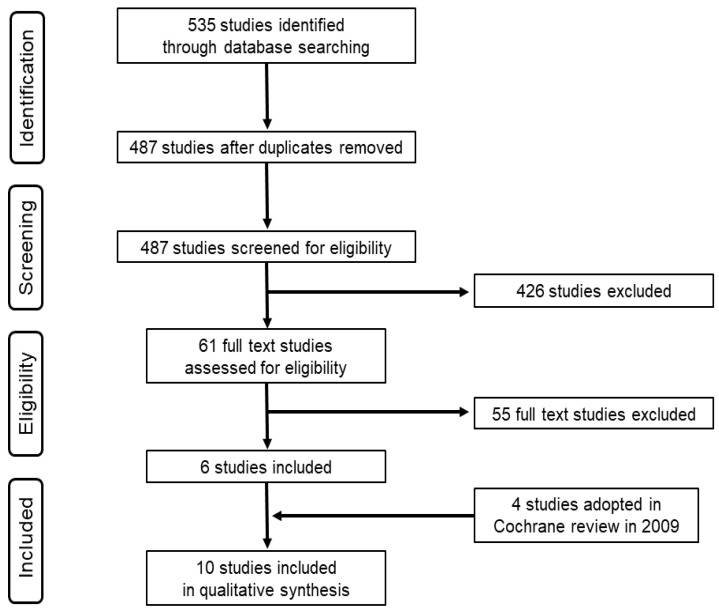
Preferred reporting items for systematic review and meta-analyses (PRISMA) flow diagram.

**Figure 2 children-10-01324-f002:**
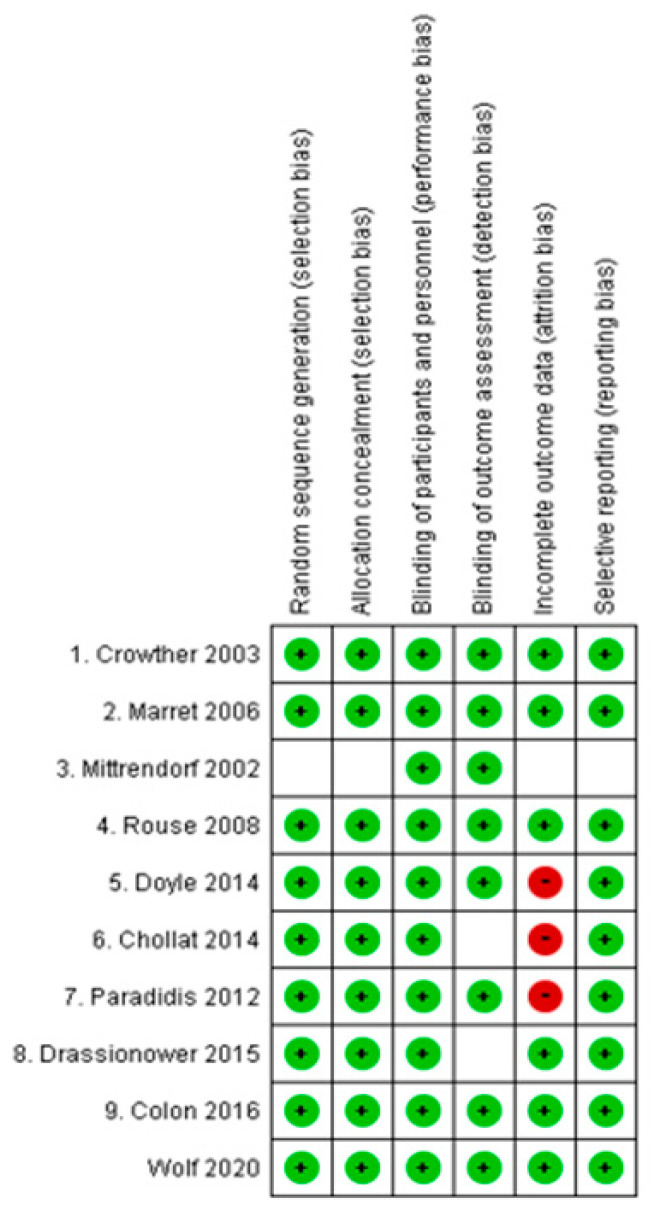
Risk of bias in the included studies [7,8,10,18,19,20,21,22,23,24].

**Figure 3 children-10-01324-f003:**
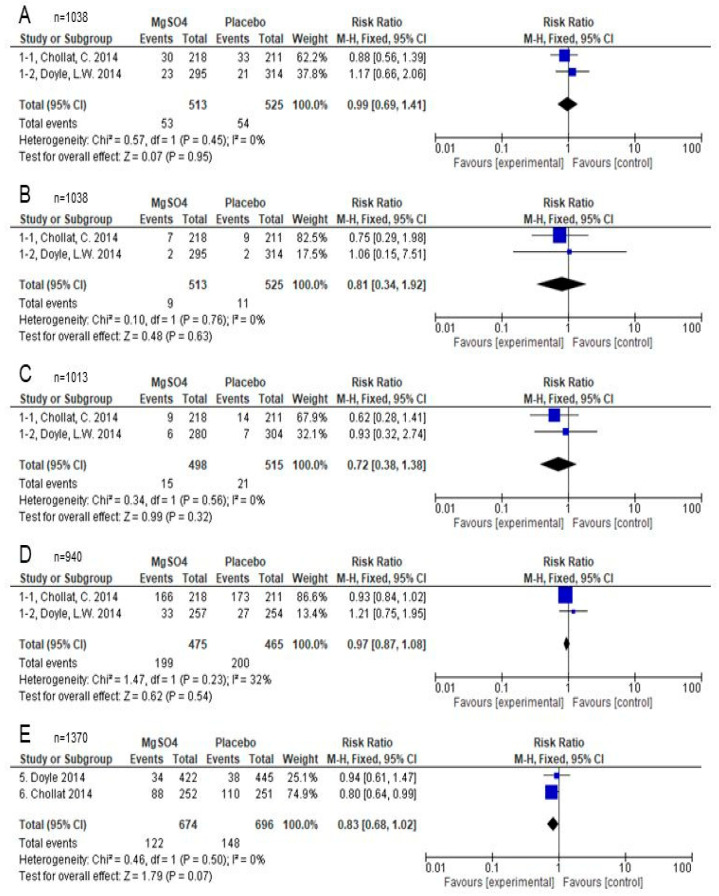
Comparison of school-age outcomes between antenatal MgSO_4_ therapy and placebo [18,19]. (**A**); cerebral palsy. (**B**); cerebral palsy (severe). (**C**); Hearing impairment. (**D**); Neurosensory disability. (**E**); Death at school-age. MgSO_4_, magnesium sulphate.

**Table 1 children-10-01324-t001:** Included studies.

No	Study	Trial Name	Participants (Infants)	School-Age Outcomes	Short-Term Outcomes	Notes
1	Crowther 2003 [7]	ACTOMgSO_4_	1255		○	Adopted in Cochrane review published in 2009
2	Marret 2006 [8]	PREMAG	688		○	Adopted in Cochrane review published in 2009
3	Mittrendorf 2002 [24]	MAGNET	59		○	Adopted in Cochrane review published in 2009
4	Rouse 2008 [10]	BEAM	2444		○	Adopted in Cochrane review published in 2009
5	Doyle 2014 [18]		669	○		School-age outcomes of No.1
6	Chollat 2014 [19]		431	○		School-age outcomes of No.2
7	Paradidis 2012 [20]		132		○	Single center analysis of No.1
8	Drassinower 2015 [21]		1047		○	Secondary analysis of No.4
9	Colon 2016 [22]		31		○	
10	Wolf 2020 [23]		680		○	

ACTOMgSO_4_; Australasian Collaborative Trial of Magnesium Sulphate; PREMAG; PREterm brain protection by MAGnesium sulphate; MAGNET; Magnesium and Neurologic Endpoints Trial; BEAM; Beneficial Effects of Antenatal Magnesium Sulphate.

**Table 2 children-10-01324-t002:** Participants.

No	Study	Participants
1	Crowther 2003 [7]	1062 women with fetuses < 30 weeks’ gestation and 1255 born infants.
2	Marret 2006 [8]	564 women with fetuses of gestational age < 33 weeks and 688 born infants.
3	Mittrendorf 2002 [24]	55 women with preterm labor at 25–33 weeks’ gestation and 59 born infants.
4	Rouse 2008 [10]	2241 women at imminent risk for delivery between 24 and 31 weeks of gestation and 2444 born infants.
5	Doyle 2014 [18]	669 infants out of 1255 infants who participated in Crowther 2003.
6	Chollat 2014 [19]	431 infants out of 688 infants who participated in Marret 2006.
7	Paradidis 2012 [20]	114 women out of 1062 women who participated in Crowther 2003 and 132 born infants.
8	Drassinower 2015 [21]	1047 singleton women out of 2241 women who participated in Rouse 2008 and 1047 born infants.
9	Colon 2016 [22]	30 women between 24 and 34 weeks of gestation presenting labor and delivery diagnosed with non-severe placental abruption and 31 born infants.
10	Wolf 2020 [23]	560 pregnant women at risk for preterm delivery before 32 weeks of gestation and 680 born infants.

**Table 3 children-10-01324-t003:** Intervention.

No	Study	Intervention
1	Crowther 2003 [7]	MgSO_4_ was intravenously administrated with a loading infusion of 4 g for 20 min followed by a maintenance infusion of 1 g/h for up to 24 h in the treatment group.
2	Marret 2006 [8]	MgSO_4_ was intravenously administrated with a loading infusion of 4 g over 30 min in the treatment group.
3	Mittrendorf 2002 [24]	MgSO_4_ was intravenously administrated with a loading infusion of 4 g in the treatment group.
4	Rouse 2008 [10]	MgSO_4_ was intravenously administrated with a loading infusion of 6 g for 20–30 min followed by a maintenance infusion of 2 g/h in the treatment group. If delivery had not occurred after 12 h and was no longer considered imminent, the infusion was discontinued and resumed when delivery was deemed imminent again.
5	Doyle 2014 [18]	Same as No.1
6	Chollat 2014 [19]	Same as No.2
7	Paradidis 2012 [20]	Same as No.1
8	Drassinower 2015 [21]	Same as No.4
9	Colon 2016 [22]	MgSO_4_ was intravenously administrated with a loading infusion of 4 g followed by a maintenance infusion of 2 g/h for up to 48 h in the treatment group. Additional 2 g boluses and rate increases of up to 4 g/h could be administrated at the discretion of the treating physician.
10	Wolf 2020 [23]	MgSO_4_ was intravenously administrated with a loading infusion of 5 g over 20–30 min, followed by a maintenance infusion of 1 g/h for up to 24 h in the treatment group.

**Table 4 children-10-01324-t004:** Long-term outcomes and certainty of evidence.

Certainty Assessment							№ of Patients		Effect		Certainty
№ of Studies	Study Design	Risk of Bias	Inconsistency	Indirectness	Imprecision	Other Considerations	MgSO_4_ and School-Age Outcomes	Placebo	Relative (95% CI)	Absolute (95% CI)	
Cerebral palsy											
2	randomized trials	serious ^a^	not serious	not serious	serious ^b^	none	53/513 (10.3%)	54/525 (10.3%)	RR 0.99 (0.69 to 1.41)	1 fewer per 1000 (from 32 fewer to 42 more)	⨁⨁○○LOW
Cerebral palsy (severe)											
2	randomized trials	not serious	not serious	not serious	very serious ^b^	none	9/513 (1.8%)	11/525 (2.1%)	RR 0.81 (0.34 to 1.92)	4 fewer per 1000 (from 14 fewer to 19 more)	⨁⨁○○LOW
Hearing impairment											
2	randomized trials	serious ^a^	not serious	not serious	serious ^b^	none	15/498(3.0%)	21/515 (4.1%)	RR 0.72 (0.38 to 1.38)	11 fewer per 1000 (from 25 fewer to 15 more)	⨁⨁○○LOW
Neurosensory disability											
2	randomized trials	serious ^a^	not serious	not serious	serious ^b^	none	199/475 (41.9%)	200/465 (43.0%)	RR 0.97 (0.87 to 1.08)	13 fewer per 1000 (from 56 fewer to 34 more)	⨁⨁○○LOW
Death at school-age											
2	randomized trials	serious ^a^	not serious	not serious	serious ^b^	none	122/981 (12.4%)	148/962 (15.4%)	RR 0.81 (0.65 to 1.01)	29 fewer per 1000 (from 54 fewer to 2 more)	⨁⨁○○LOW
Maternal adverse events (any symptom)											
3	randomized trials	not serious	serious ^c^	not serious	serious ^b^	none	1311/1628 (80.5%)	342/1667 (20.5%)	RR 2.79 (1.10 to 7.05)	367 more per 1000 (from 21 more to 1000 more)	⨁⨁○○LOW

CI: Confidence interval; RR: Risk ratio; Explanations; ^a^ Attrition bias was high; ^b^ Wide 95%CI crossing non-significant line; ^c^ I-square is over 85%. High heterogeneity. The degree of certainty was expressed as the number of ⊕.

## Data Availability

Not applicable.

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
