# Peer review of "School-Age Outcomes of Antenatal Magnesium Sulphate in Preterm Infants"

_children, 2023, doi:10.3390/children10081324_

Round 1

Reviewer 1 Report

This is a systematic review of the effects of antenatal MgSO4 on outcomes for preterm infants and their mothers. As far as I’m aware, it’s the first such review on long-term outcomes: it reviews RCTs that follow these children into their school years.

Previous systematic reviews (including a well-known Cochrane review from 2009) have reported that antenatal MgSO4 reduces the risk of cerebral palsy (CP). The present review, however, reports that their analysis of school-age data shows that “antenatal MgSO4 therapy had no impact on cerebral palsy”. I have some concerns about CP being treated as a long-term outcome because, as the authors themselves say, “CP is unlikely to be diagnosed after 2 years of age newly.” Nor does CP disappear once it has been diagnosed. So the RCTs with short-term outcomes should be sufficient to answer that question.

The conclusions drawn in this paper for the incidence of CP in pretermers whose mothers did or did not receive MgSO4 prenatally, therefore, seem to me not based on the strongest available. Only 2 of the original studies (which happened to be ones with weak effects originally) could be included in the present analysis, and it is noteworthy that the dropout rates in these 2 studies were so high that only about half of the original participants were followed up into their school years. According to Supporting Information 1 in this paper, the conclusion about a lack of difference between MgSO4 and control groups in rates of CP was based on only 1038/1943 = 53.4% original participants.

Therefore, if I want to know whether antenatal MgSO4 reduces the prevalence of CP, my best source of evidence would be from data collected in infancy (short-term data), in which 100% of participants are included in the analysis. 

I think that the authors of the present systematic review are aware of all this. And it is to their credit that they have been so clear and transparent in presenting the results, carefully noting the dropout rates in the 2 studies of school-age outcomes. However, I think that their conclusions in text need to be qualified, both in the Discussion and in the Abstract. It is not really true to say that “antenatal MgSO4 therapy had no impact on cerebral palsy.” The best evidence from infant studies shows that antenatal MgSO4 therapy does have an impact on cerebral palsy; but long-term data from school-age populations is currently insufficient to show the same because most of the RCTs haven’t reported it, and those that have reported it have nearly 50% missing data.

If the data on CP is so questionable, then this also raises questions about the conclusion based on the other school-age data, because this is also based on only 2 of the RCTs with nearly 50% missing data.

Apart from this major recommendation regarding the interpretation of results, I have some minor ones, as follows:

1. In accordance with PRISMA, item 7, please present the full search strategies for all databases. Include the manual supplemental searches that the reviewers conducted.

2. Define the abbreviation RR. Is it risk ratio?

3. “I2” should be either “I square” or “I” with a superscripted 2.

4. Change “We assessed seven domains such as random sequence” to “We assessed seven domains: random sequence”—as you have listed all 7 domains in the remainder of the sentence.

5. The first paragraph on page 5 would seem to belong better in a table than in text.

6. Table 3: Please give a more precise title, such as “Long-term outcomes and certainly of evidence”.

7. Figure 3 looks slightly blurry. Can you sharpen the picture quality?  

8. Some of the sections in text under 3.8 become repetitive. Would these results go better into a table, with only a summarizing statement in text?

9. Change “By the way, the other items” to “The other items”.

10. This sentence is unclear and not grammatical: “In addition, dropout rate of two studies was high, although there was not so difference between follow-up group and dropout group.”

Only minor changes needed: see review. 

Author Response

Thank you very much for providing important insights. In the following sections, you will find our responses to each of your points and suggestions. We have noted in red the corrections in the manuscript. We are grateful for the time and energy you expended on our behalf.

This is a systematic review of the effects of antenatal MgSO4 on outcomes for preterm infants and their mothers. As far as I’m aware, it’s the first such review on long-term outcomes: it reviews RCTs that follow these children into their school years.

Previous systematic reviews (including a well-known Cochrane review from 2009) have reported that antenatal MgSO4 reduces the risk of cerebral palsy (CP). The present review, however, reports that their analysis of school-age data shows that “antenatal MgSO4 therapy had no impact on cerebral palsy”. I have some concerns about CP being treated as a long-term outcome because, as the authors themselves say, “CP is unlikely to be diagnosed after 2 years of age newly.” Nor does CP disappear once it has been diagnosed. So the RCTs with short-term outcomes should be sufficient to answer that question.

The conclusions drawn in this paper for the incidence of CP in pretermers whose mothers did or did not receive MgSO4 prenatally, therefore, seem to me not based on the strongest available. Only 2 of the original studies (which happened to be ones with weak effects originally) could be included in the present analysis, and it is noteworthy that the dropout rates in these 2 studies were so high that only about half of the original participants were followed up into their school years. According to Supporting Information 1 in this paper, the conclusion about a lack of difference between MgSO4 and control groups in rates of CP was based on only 1038/1943 = 53.4% original participants.

Therefore, if I want to know whether antenatal MgSO4 reduces the prevalence of CP, my best source of evidence would be from data collected in infancy (short-term data), in which 100% of participants are included in the analysis. 

I think that the authors of the present systematic review are aware of all this. And it is to their credit that they have been so clear and transparent in presenting the results, carefully noting the dropout rates in the 2 studies of school-age outcomes. However, I think that their conclusions in text need to be qualified, both in the Discussion and in the Abstract. It is not really true to say that “antenatal MgSO4 therapy had no impact on cerebral palsy.” The best evidence from infant studies shows that antenatal MgSO4 therapy does have an impact on cerebral palsy; but long-term data from school-age populations is currently insufficient to show the same because most of the RCTs haven’t reported it, and those that have reported it have nearly 50% missing data.

Answer: Thank you very much for your important remarks. We also have been feeling some discomfort with this conclusion. We wondered if we could really conclude that “antenatal MgSO4 therapy had no impact on cerebral palsy", although we had only about half of data. However, the review texts you pointed out convinced me very much. As the reviewer indicated, we also think that conclusions based on data from infancy, which has a high follow-up rate, are more reliable. Therefore, we have changed our conclusions in the abstract and discussion (P1 Line19, 25-27, P11 Line335-345, P12 Line385, 392-393). We appreciate you so much.

If the data on CP is so questionable, then this also raises questions about the conclusion based on the other school-age data, because this is also based on only 2 of the RCTs with nearly 50% missing data.

Apart from this major recommendation regarding the interpretation of results, I have some minor ones, as follows:

  1. In accordance with PRISMA, item 7, please present the full search strategies for all databases. Include the manual supplemental searches that the reviewers conducted.

Answer: Full search strategies for all databases were provided in Supporting information 1. We conducted the manual supplemental searches in only MEDLINE (P2 Line69-70).

  1. Define the abbreviation RR. Is it risk ratio?

Answer: As the reviewer indicated, RR means risk ratio. We described the definition of RR (P1 Line42).

  1. “I2” should be either “I square” or “I” with a superscripted 2

Answer: We have changed to “I” with a superscripted 2.

  1. Change “We assessed seven domains such as random sequence” to “We assessed seven domains: random sequence”—as you have listed all 7 domains in the remainder of the sentence.

Answer: We have changed the relevant section (P3 Line107).

  1. The first paragraph on page 5 would seem to belong better in a table than in text.

Answer: We have changed the relevant section to Table 2.

  1. Table 3: Please give a more precise title, such as “Long-term outcomes and certainly of evidence”.

Answer: We have changed the title of Table 4.

  1. Figure 3 looks slightly blurry. Can you sharpen the picture quality?  

Answer: We have tried to sharpen the picture quality of Figure 3.

  1. Some of the sections in text under 3.8 become repetitive. Would these results go better into a table, with only a summarizing statement in text?

Answer: We tried to, but could not produce a useful table, because some data overlapped with Table 4, Figure 3, and Supporting information 4. We hope you understand.

  1. Change “By the way, the other items” to “The other items”.

Answer: We have changed the relevant section (P11 Line366).

  1. This sentence is unclear and not grammatical: “In addition, dropout rate of two studies was high, although there was not so difference between follow-up group and dropout group.”

Answer: We have removed the relevant section (P12 Line375).

Reviewer 2 Report

This is a systematic review and meta-analysis aimed at evaluating the effects of antenatal MgSO4 therapy on school-age outcomes of preterm infants. The authors analyzed the available data from 10 randomized controlled trials, only two of them were on school-age outcomes. They indicated that the antenatal MgSO4 therapy had no impact on cerebral palsy, hearing impairment, neurosensory disability, and death at school age. However, it increased the risk of maternal adverse events in the short term, but without neonatal symptoms. The review deals with an important topic for clinicians, however, there exist some concerns/suggestions that the authors need to address. Please find them below.

1.       The authors may need to provide further context and background information about the role of antenatal MgSO4 therapy (in light of the available evidence) on the short and long-term neurological outcomes to contextualize the review.

2.       The study's justification is obscure. The gaps in the body of literature should be pointed out by authors, who should also underline how critical it is to fill such gaps. Furthermore, a stronger justification would also result in a declaration of the potential implications for practice, which is currently missing.

3.       Could authors demonstrate more clearly what has been done to ensure explicit and reproducible criteria to select articles that are eventually included in the review (i.e., a detailed definition of the inclusion criteria would be interesting)?

4.       Have you looked at a particular section of articles (titles, abstracts, table of contents) during the search and collection stage, or have you excluded this large number of studies after reading the full-text articles?

5.       When there was a potential source of disagreement between reviewers to decide on study inclusion, the disagreements were resolved by discussion or by consultation with an adjudicator. How they reached a consensus? The authors should elaborate more on this point in the methodology.

6.       Are results from 10 studies included in the meta-analysis (only 2 of which were on school-age outcome)  enough to synthesize the current evidence on the effects of antenatal MgSO4 therapy on school-age outcomes of preterm infants?

7.       The discussion reads well. However, it would benefit from further comparison with previous review findings or general literature.

8.       Some references (specifically # 3, 4, 5, 6, and 7) are fairly outdated. Could the authors substitute them with more updated citations?

Author Response

Thank you very much for providing important insights. In the following sections, you will find our responses to each of your points and suggestions. We have noted in red the corrections in the manuscript. We are grateful for the time and energy you expended on our behalf.

This is a systematic review and meta-analysis aimed at evaluating the effects of antenatal MgSO4 therapy on school-age outcomes of preterm infants. The authors analyzed the available data from 10 randomized controlled trials, only two of them were on school-age outcomes. They indicated that the antenatal MgSO4 therapy had no impact on cerebral palsy, hearing impairment, neurosensory disability, and death at school age. However, it increased the risk of maternal adverse events in the short term, but without neonatal symptoms. The review deals with an important topic for clinicians, however, there exist some concerns/suggestions that the authors need to address. Please find them below.

  1. The authors may need to provide further context and background information about the role of antenatal MgSO4 therapy (in light of the available evidence) on the short and long-term neurological outcomes to contextualize the review.

Answer: We have added more information in the manuscript (P1-2 Line45-46).

  1. The study's justification is obscure. The gaps in the body of literature should be pointed out by authors, who should also underline how critical it is to fill such gaps. Furthermore, a stronger justification would also result in a declaration of the potential implications for practice, which is currently missing.

Answer: Thank you for your important remarks. As the reviewer pointed out, it is not possible to make an accurate evaluation of long-term outcome in the current situation where only about half of data is available after school-age. Therefore, we have changed our conclusion regarding cerebral palsy, and have described that it is particularly important to accumulate long-term outcome data including school-age in the future (P11 Line335-345).

  1. Could authors demonstrate more clearly what has been done to ensure explicit and reproducible criteria to select articles that are eventually included in the review (i.e., a detailed definition of the inclusion criteria would be interesting)?

Answer: We have described the process in more detail how we selected the review articles (P2 Line72-79). In addition, we made Supporting information 1 which presented full search strategy for all databases.

  1. Have you looked at a particular section of articles (titles, abstracts, table of contents) during the search and collection stage, or have you excluded this large number of studies after reading the full-text articles?

Answer: First, we checked title and abstract of articles and excluded articles we decided unnecessary. Second, we read the full-text articles we had adopted and excluded articles we decided unnecessary. We have described the process (P2 Line74-76).

  1. When there was a potential source of disagreement between reviewers to decide on study inclusion, the disagreements were resolved by discussion or by consultation with an adjudicator. How they reached a consensus? The authors should elaborate more on this point in the methodology.

Answer: If there was a disagreement between two reviewers during this process, each presented the reasons for inclusion or exclusion and resolved through discussion. In most cases, consensus was reached during the discussion. Only when consensus was not reached, we consulted with a third evaluator and followed his decision. We have described the process (P2 Line76-79).

  1. Are results from 10 studies included in the meta-analysis (only 2 of which were on school-age outcome) enough to synthesize the current evidence on the effects of antenatal MgSO4 therapy on school-age outcomes of preterm infants?

Answer: Thank you for your important remarks. As the reviewer pointed out, we don't think it is enough, because only about half of the original participants were followed up to school-age. Especially for cerebral palsy (CP), there are few new cases diagnosed after early childhood and once is diagnosed, it is not reversed. So we decided that conclusions from data of early childhood, which has a high follow-up rate, are more reliable. Therefore, we have changed our conclusion regarding CP. We also emphasized the need for studies that include long-term data to accurately assess school-age outcomes (P1 Line19, 25-27, P11 Line335-345, P12 385, 392-393).

  1. The discussion reads well. However, it would benefit from further comparison with previous review findings or general literature.

Answer: We have added more information such as comparisons with previous review findings (P11-12 Line357-362, 365-366, 369-373).

  1. Some references (specifically # 3, 4, 5, 6, and 7) are fairly outdated. Could the authors substitute them with more updated citations?

Answer: We have substituted some references with more updated articles.

Round 2

Reviewer 1 Report

This is the second round of review for this paper. It is a systematic review of the long-term (school-age) outcomes of antenatal MgSO4 for preterm infants and their mothers.

In my earlier review, my main concern was that the conclusion of this review (antenatal MgSO4 therapy had no impact on cerebral palsy) was unwarranted, as there were no follow-up data for nearly half the participants. The authors have responded by changing their conclusions to say that there is insufficient evidence of long-term outcomes owing to the large percentage of missing data. I am entirely happy with this revised version of the paper.

I have a few minor suggestions to make on this version:

In the title to Table 4, “certainly” should be “certainty”. Sorry—this was my error in my previous review.

“The reason would be because only about half of the original participants were followed up to school-age.” I suggest changing this to “The reason would be because only 1038/1943 = 53.4% of the original participants were followed up to school-age.” (I think it is more convincing if you give a specific fraction.)

“However, we had no school-age data of Rouse 2008, too. Regarding CP, it is unlikely to be diagnosed after 2 years of age newly and do not disappear once it has been diagnosed. And conclusion from data collected in early childhood, which had a high follow-up rate, would be more reliable. Therefore, the best evidence from infant studies shows that ante-natal MgSO4 therapy have an impact on cerebral palsy, but long-term data from school-age populations was insufficient to show the same evidence. Accumulation of long-term data including school-age is strongly required.” There are a few minor changes here to make the English more natural. I suggest changing it to ““However, we had no school-age data from Rouse 2008. CP is unlikely to be diagnosed after 2 years of age and does not disappear once it has been diagnosed. Therefore, it is more reliable to draw conclusions from data collected in early childhood, which had a higher follow-up rate. The best evidence from infant studies shows that ante-natal MgSO4 therapy has an impact on CP, but long-term data from school-age populations were insufficient to show the same. Accumulation of long-term data including school-age is strongly needed.”

Change “by protecting fetal brain” to “by protecting the fetal brain

Reviewer 2 Report

The authors addressed all concerns satisfactorily. I have no further comments.

Author Response

Nothing.